# Infrared nanospectroscopy reveals the molecular interaction fingerprint of an aggregation inhibitor with single Aβ42 oligomers

Francesco Simone Ruggeri [1,2 ✉], Johnny Habchi[1], Sean Chia[1], Robert I. Horne [1], Michele Vendruscolo [1 ✉] & Tuomas P. J. Knowles [1,3 ✉]

Significant efforts have been devoted in the last twenty years to developing compounds that can interfere with the aggregation pathways of proteins related to misfolding disorders, including Alzheimer's and Parkinson's diseases. However, no disease-modifying drug has become available for clinical use to date for these conditions. One of the main reasons for this failure is the incomplete knowledge of the molecular mechanisms underlying the process by which small molecules interact with protein aggregates and interfere with their aggregation pathways. Here, we leverage the single molecule morphological and chemical sensitivity of infrared nanospectroscopy to provide the first direct measurement of the structure and interaction between single Aβ42 oligomeric and fibrillar species and an aggregation inhibitor, bexarotene, which is able to prevent Aβ42 aggregation in vitro and reverses its neurotoxicity in cell and animal models of Alzheimer's disease. Our results demonstrate that the carboxyl group of this compound interacts with Aβ42 aggregates through a single hydrogen bond. These results establish infrared nanospectroscopy as a powerful tool in structure-based drug discovery for protein misfolding diseases.

[1] Department of Chemistry, Center for Misfolding Diseases, University of Cambridge, Cambridge, UK. [2] Laboratory of Organic Chemistry & Laboratory of Physical Chemistry, Wageningen University, Wageningen, The Netherlands. [3] Cavendish Laboratory, University of Cambridge, Cambridge, UK. ✉email: simone.ruggeri@wur.nl; mv245@cam.ac.uk; tpjk2@cam.ac.uk

Alzheimer's disease (AD) is characterised by memory loss and cognitive impairment. AD is the primary cause of dementia, which affects currently over 50 million people worldwide, a number expected to exceed 150 million by 2050 (refs. [1–3]). The self-assembly of the 42-residue isoform of the amyloid-β (Aβ) peptide into intractable aggregates is considered to be at the core of the molecular pathways leading to AD[3–5]. It is therefore important to understand at the molecular level the aggregation process of Aβ42 in order to develop effective therapeutic strategies that aim at inhibiting its self-assembly.

Great efforts have been devoted in the last 20 years to understand the molecular basis of this devastating disorder and to develop small molecules that could interfere with the aggregation pathway of Aβ42 (refs. [6–12]). Indeed, disease-modifying small molecules represent c.a. one-third of the registered trials, in which anti-Aβ therapies dominate[13]. However, despite these efforts, no compound has entered the clinical use to date[13,14].

These repeated failures are in part related to an incomplete understanding of the molecular mechanisms underlying the process by which small molecules interact with protein aggregates and how they interfere with the pathways of aggregation. It is increasingly recognised that inhibiting Aβ aggregation per se could have unexpected consequences on the toxicity, as it could not only decrease it but also leave it unaffected, or even increase it[15]. This complexity is due to the non-linear nature of the aggregation network, in which neurotoxic small oligomeric intermediates[16–20] can be formed in a variety of ways, some of which highly sensitive to small perturbation. Therefore, promising effective therapeutic strategies must be aimed at targeting precise species in a controlled intervention during the aggregation process.

We recently described a drug discovery strategy based on the accurate characterisation of the effects of small molecules on the kinetics of Aβ aggregation[21]. This strategy exploits the power of chemical kinetics[22,23], in which the effect of small molecules on the rates of specific microscopic steps in Aβ42 aggregation can be determined quantitatively. Using this strategy, we identified an FDA-approved anti-cancer drug, bexarotene, which was found to delay substantially the aggregation of Aβ42 and to reduce the associated toxicity in vivo[21,24]. Bexarotene delays the aggregation of Aβ42 in a concentration-dependent manner by inhibiting primary and secondary pathways[21]. Although a kinetic understanding of the molecular mechanism by which small molecules affect Aβ aggregation has been achieved, the chemical interactions and the structural information on Aβ species in the free and bound forms to the small molecule remain to be unravelled. This limitation is mainly due to the lack of experimental tools that allow accurate direct measurements of the chemical and structural properties of individual species in a heterogeneous population.

A breakthrough in the analysis of the chemical properties of heterogeneous protein aggregation at the nanoscale has been achieved with the development and application of atomic force microscopy in combination with infrared nanospectroscopy (AFM-IR; Fig. 1d)[25–35]. AFM-IR is becoming widely applicable in biology since is capable of acquiring simultaneously morphological, nanomechanical and nanoscale resolved chemical IR maps and absorption spectra from protein aggregates, liquid–liquid phase separated condensates, chromosomes and single cells[30,32,35–37]. As fundamental achievement, we have recently demonstrated that ORS-nanoIR enables to acquire IR absorption spectra for secondary structure determination from single protein molecules[38].

In the present work, we exploit the powerful combination of single-molecule imaging and vibrational spectroscopy offered by AFM-IR to characterise at the single aggregate scale the morphological, chemical and structural properties of the aggregated species of Aβ42 in the absence and the presence of bexarotene. We thus provide a direct measurement at the single oligomeric scale of the interactions between a small molecule and Aβ42 oligomers and fibrils. Our results constitute a key finding in the investigation of the transient and heterogeneous conformational changes occurring during protein aggregation as well as the capability to study the interactions between proteins and therapeutic compounds, with the aim of developing therapeutics against neurodegenerative disorders.

## Results

**Bexarotene inhibits Aβ42 aggregation.** Before investigating the chemical interactions between bexarotene and Aβ42 aggregates at the nanoscale, we carried out a characterisation of the aggregation process of Aβ42 (Fig. 1). As shown by phase-controlled high-resolution AFM imaging, before incubation at 0 h, Aβ42 is largely in a monomeric state and there are no fibrillar aggregates visible (Fig. 1a). After 4 h incubation, as previously reported, we observe an abundance of fibrillar aggregates with typical cross-sectional diameters between 2 and 6 nm (Supplementary Fig. 1). We further performed a kinetic analysis of the aggregation of Aβ42 using a highly reproducible ThT-based protocol[39]. In particular, we investigated the aggregation reaction in the absence and the presence of threefold excess of bexarotene and one of its derivatives where the carboxyl group was substituted by an ester functional group, leading to the absence of an acidic ionisable group (Fig. 1c and Supplementary Figs. 2 and 3). The rationale behind the choice of this derivative is based on our previous experiments that have systematically assessed how different modifications of the hydrophilic and hydrophobic parts of the bexarotene scaffold affect the kinetics of aggregation of Aβ42[40]. The aggregation displays the three phases of a typical nucleation-polymerisation reaction: lag, growth and plateau. Indeed, typical amyloid formation displays sigmoidal growth kinetics, where a transition zone, namely the growth phase, is preceded and followed by flat regions, usually referred to as the lag phase and the plateau, respectively. Bexarotene delayed the aggregation of Aβ42 significantly, by inhibiting both primary and surface-catalysed secondary nucleation as previously demonstrated[21]. On the other hand, its derivative did not affect the aggregation reaction even at high concentrations. These results show that bexarotene can slow down the kinetics of aggregation reaction through a direct interaction with Aβ42 aggregates.

**Infrared nanospectroscopy of single Aβ42 oligomers and fibrils.** We first applied conventional bulk IR spectroscopy to investigate the chemical binding between Aβ42 fibrils formed at the end of the aggregation reaction (plateau phase, 4 h) and bexarotene (Supplementary Fig. 4). However, with bulk IR spectroscopy, we were not able to observe any difference between the fibrils formed in presence and absence of bexarotene, and thus we could not study their chemical binding using conventional FTIR. This limitation is largely related to the nature of this bulk technique, which is not able to discriminate between bound small molecules and the ones diffused in solution.

To overcome these limitations, we next exploited the nanoscale chemical resolution power of AFM-IR to study the chemical and structural properties of Aβ42 aggregates at the single molecule scale (Fig. 1d–e) in the absence and presence of bexarotene. Consequently, ThT-free aliquots of Aβ42 solutions were removed at different time points from the aggregation reactions for single-molecule measurements. In particular, we focused on the growth phase around the half-time of the reactions (2 h for Aβ42 alone and 4.5 h for Aβ42+bexarotene) and on the plateau phase of

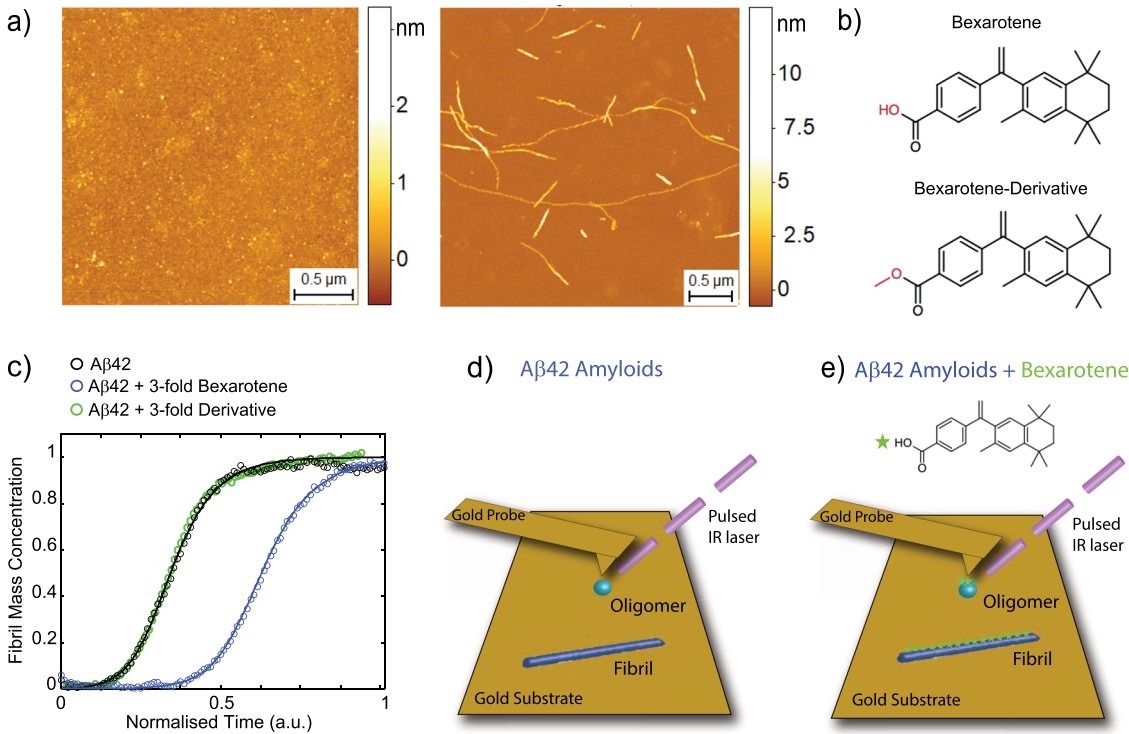

**Fig. 1 Effect of bexarotene on the aggregation kinetics of Aβ42. a** Representative high-resolution AFM morphology maps of Aβ42 aggregates before incubation (0 h) and at the end of the aggregation reaction (4 h). The morphology of the aggregates was reproducible and consistent on at least five independent and randomly chosen areas on the surface of each sample. **b** Structures of bexarotene and its derivative, where the hydroxyl group is replaced with a methyl group (highlighted in red). **c** Kinetics profiles of the aggregation reaction of 2 μM Aβ42 in the absence or in the presence of a 3:1 concentration ratio of bexarotene or its derivative. The solid lines show the best fit of the experimental data (open circles) when primary and secondary pathways are both inhibited by bexarotene. **d**, **e** Schematic of the AFM-IR experimental approach to study the interaction of Aβ42 oligomers (light blue sphere) and fibrils (dark blue rod) with bexarotene (green star) at the single aggregate scale. To reach high sensitivity off-resonance, small pulse ORS-nanoIR at the nanogap between a gold-coated probe and a gold substrate was used[38].

aggregation after 15 h incubation. We deposited the aliquots of Aβ42 aggregates on atomically flat (0.4 nm RMS) template gold surfaces. As expected, in the growth phase we observed the co-existence of oligomeric and fibrillar aggregates, while in the plateau phase we observed only fibrillar aggregates (Fig. 2a, b and Supplementary Fig. 5), similarly as observed for samples deposited on conventional mica substrates (Fig. 1).

Then, we focused on the measurement of the single aggregate chemical and structural properties of Aβ42 aggregates alone. The sensitivity of AFM-IR has been recently demonstrated to be capable to reach the detection of single protein molecules of apoferritin with a molecular weight of ~400 kDa and hydro-dynamic radius of ~12 nm, with high signal-to-noise ratio in order to determine their secondary structure[38]. This high sensitivity was reached by measuring at a short laser pulse, off-resonance (ORS-nanoIR)[38] and exploiting the field enhancement at the nanogap with the metallic probe and substrate and reach single protein molecule sensitivity. We applied similar principles in the present study to reach the detection of the chemical signature of individual oligomeric (Fig. 2c) and fibrillar species of Aβ42 (Fig. 2d) with a typical cross-sectional diameter of both oligomeric and fibrillar ranging between ~3 and 8 nm (Supplementary Fig. 6). To exclude from our chemical and structural analysis the residual absorption of contaminants on the tip and on the substrate, we subtracted each spectrum acquired on one aggregate from the residual absorption on the substrate (Supplementary Fig. 7).

We then applied second derivative analysis to deconvolve the amide band I of the aggregates and evaluate the structural

contributions to their secondary and quaternary structure (Fig. 2e, f). The oligomeric aggregates presented their major structural contribution at 1648 cm$^{-1}$, indicating the prevalence of random coil conformation, but also a significant peak at 1695 cm$^{-1}$ indicating the presence of intermolecular antiparallel β-sheets (Fig. 2e). In the case of the fibrillar aggregates, we could observe the presence of both a high frequency band (1695 cm$^{-1}$) and a low-frequency band (1630 cm$^{-1}$) associated with intermolecular β-sheet (Fig. 2f), which in the case of Aβ42 together indicate a structural shift to a parallel β-sheet arrangement[41–44]. These results demonstrate at the single aggregate scale the conversion step from an antiparallel β-sheet conformation in oligomeric species to a parallel β-sheet conformation in the fibrillar products of the aggregation kinetics[45]. In the case of the oligomers, their cross-sectional diameter (~5–8 nm) and volume was below the size of a single protein molecule of apoferritin (~12 nm), which was only very recently achieved[38]. Although we could reach this high sensitivity, the measurement of the spectra of the oligomers was at the limit of the experimental sensitivity leading to a higher experimental noise and a partial suppression of the amide band II, as observed for single proteins previously[38].

**Interaction of single oligomers and fibrils with bexarotene**. We next applied AFM-IR to unravel the chemical interaction and effect of bexarotene on the aggregated species of Aβ (Fig. 3). We acquired first AFM morphology maps of single oligomeric (Fig. 3a) and fibrillar (Fig. 3b) species formed during the kinetics of aggregation in the presence of bexarotene. The aggregates showed similar morphology and cross-sectional dimensions as

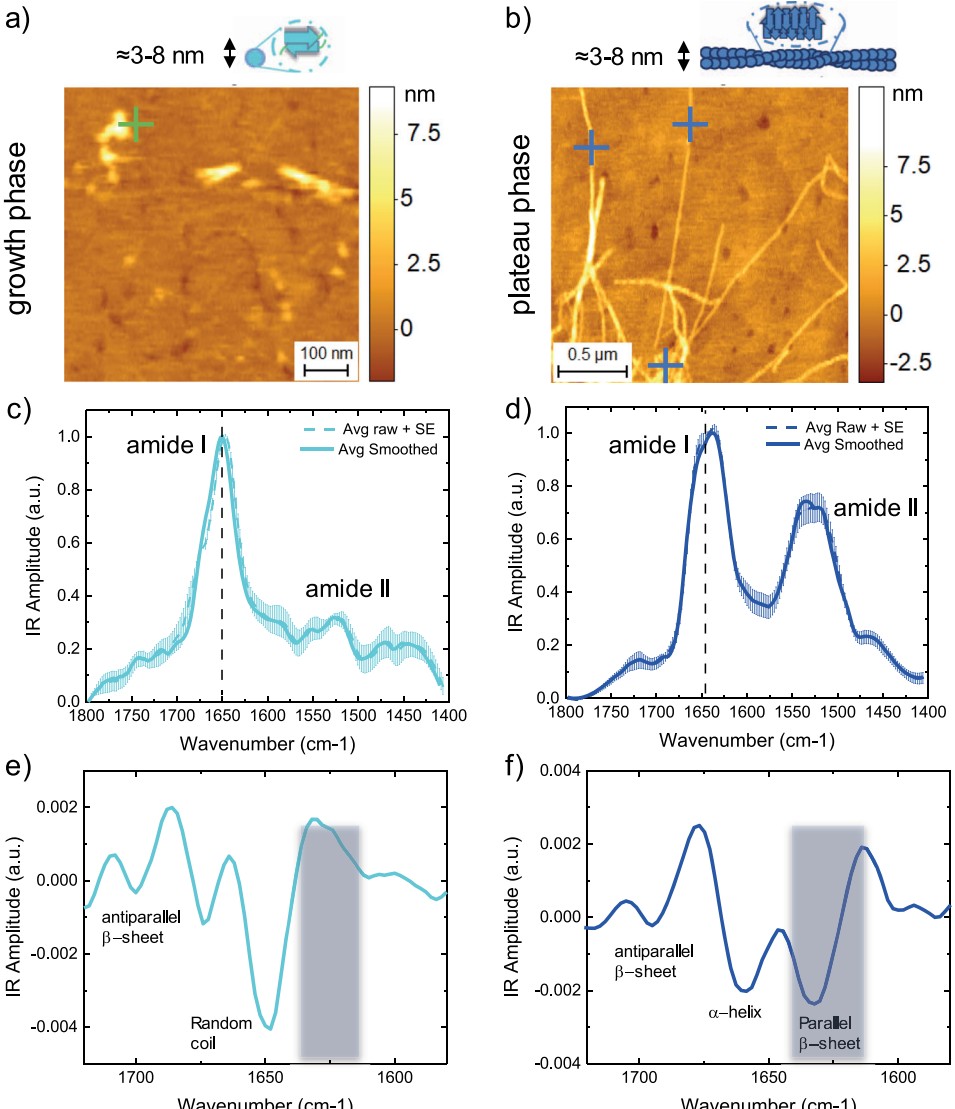

**Fig. 2 Single molecule measurements of secondary and quaternary structure of Aβ42 aggregates. a, b** AFM-IR morphology maps of single oligomeric (light blue sphere) (**a**) and fibrillar species (dark blue rod) (**b**) of Aβ42. The morphology of the aggregates was reproducible and consistent on at least five independent and randomly chosen areas on the surface of each sample. **c, d** Nanoscale localised IR absorption spectra of oligomeric (**c**) and fibrillar species (**d**) of Aβ42. We represent two spectra: (i) the average + SE of the raw spectra and (ii) the average + SE of the preprocessed spectra by applying an adjacent averaging filter (5 pts) and a Savitzky–Golay filter (second order, 15 pts). At least five different raw spectra at each position (coloured crosses) were acquired. **e, f** The second derivatives of the spectra are calculated with a Savitzky–Golay filter (second order, 15 pts) to deconvolve the major secondary structural contributions of the oligomeric and fibrillar species.

the ones in absence of the small molecule (Supplementary Fig. 5). Then, we acquired nanoscale resolved spectra from both oligomeric and fibrillar species incubated in the presence of bexarotene (Fig. 3c, d). The spectra of the aggregates in the presence of bexarotene showed marked differences when compared to the spectra of the aggregates incubated in the absence of the small molecule. In particular, we observed a significantly increased absorption between 1750–1700 and 1425–1475 $cm^{-1}$ (green boxes in Fig. 3c, d). The nanoscale localised spectra showed only minor differences in the region of the amide I band (1700–1600 $cm^{-1}$), thus indicating that the fibrillar aggregates unbound and bound to bexarotene had similar secondary structure, in good agreement with the morphological observations (Supplementary Fig. 6). In order to prove that the spectral differences measured between aggregated species in the presence and absence of

bexarotene were statistically significant, we further performed a principal components analysis (PCA; Fig. 3e–g).

PCA allowed noise reduction and the detection of subgroupings within the spectra of the aggregates in the presence and absence of bexarotene. The score plots of the analysis (Fig. 3e, f) represent each single nanoscale resolved IR spectrum as a point in the multidimensional space of principal components (PCs). The first three PCs were sufficient to represent more than 85% of the spectra variability in the ensemble and demonstrate the high statistical significance of the subgrouping (Fig. 3e) of the spectroscopic signature of the aggregates formed in the absence (blue) and presence (red) of bexarotene. The spectral differences represented by each principal component are shown in the loading plots (Fig. 4g and Supplementary Fig. 8), which show the spectroscopic region responsible for the greatest degree of

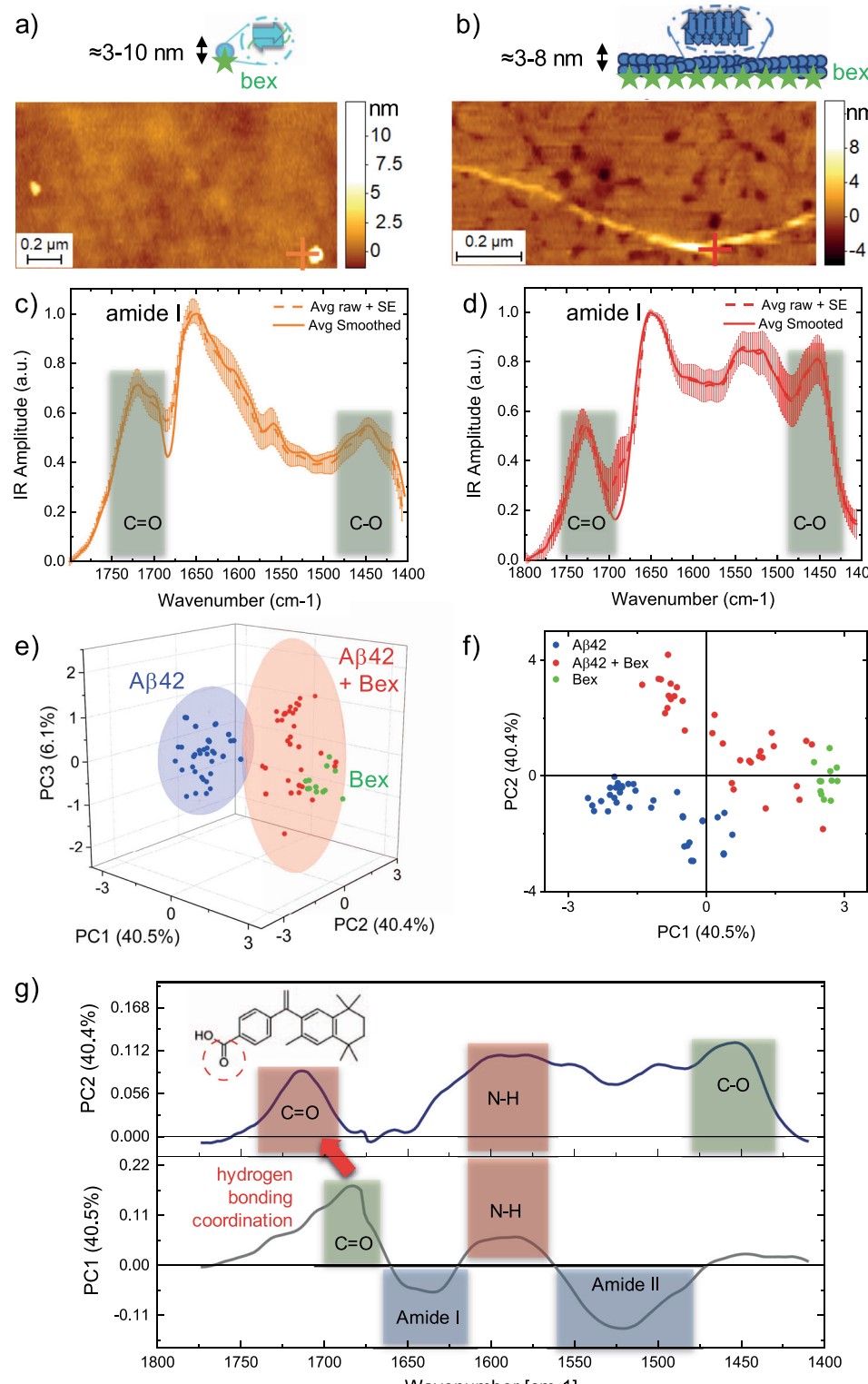

**Fig. 3 Structural characterisation of the interaction of bexarotene with single Aβ42 oligomers and fibrils.** AFM-IR morphology maps **a** oligomeric (light blue sphere) and **b** fibrillar species (dark blue rod). The morphology of the aggregates was reproducible and consistent on at least five independent and randomly chosen areas on the surface. Nanoscale localised IR absorption spectra (**c**, **d**) of oligomeric (**a**, **c**) and fibrillar species (**b**, **d**) of Aβ42 incubated with bexarotene (Bex). We represent two spectra: (i) the average + SE of the raw spectra and (ii) the average + SE of the preprocessed spectra by applying an adjacent averaging filter (5 pts) and a Savitzky–Golay filter (second order, 15 pts). At least five different raw spectra at each position (coloured crosses) were acquired. **e**, **f** 3-D and 2-D score plots of PCA analysis applied to Aβ42 aggregates (oligomers and fibrils, $n = 40$) in the absence of bexarotene (blue), Aβ42 aggregates in the presence of bexarotene (oligomers and fibrils, $n = 31$, red) and bexarotene alone ($n = 9$, green). The coloured ellipsoids represent the 90% confidence of the variance of each group, thus demonstrating the separation of the spectroscopic signature of the aggregates in presence and absence of bexarotene. **g** Loading plots of the PC1 and PC2.

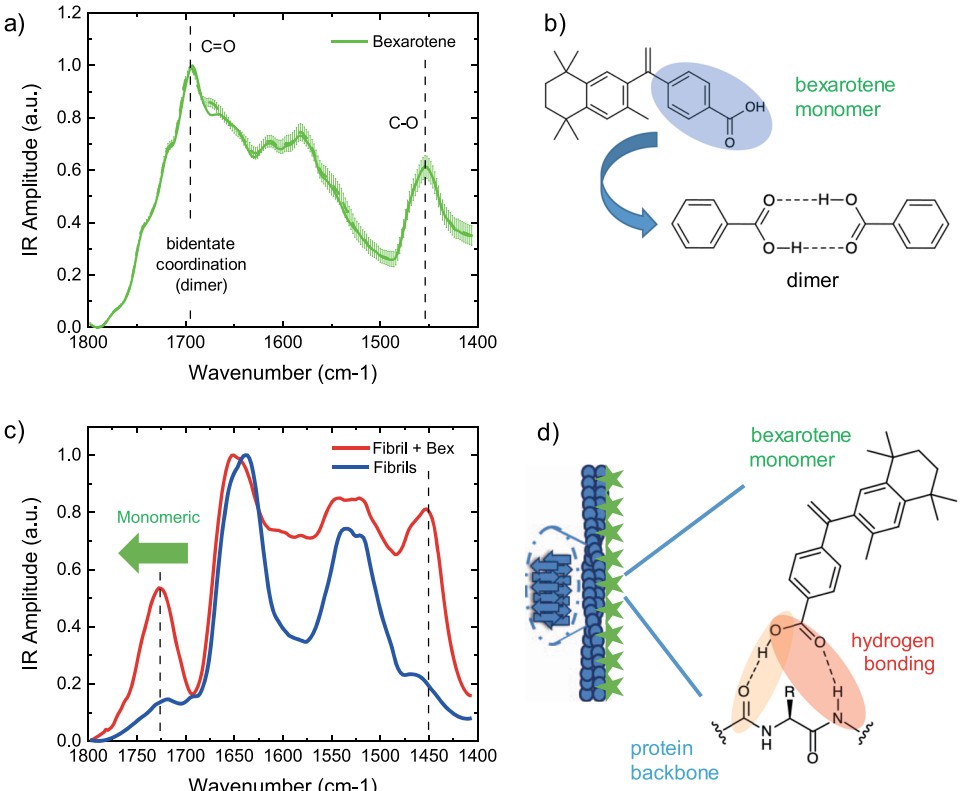

**Fig. 4 Bexarotene interacts with Aβ42 aggregates through hydrogen bonding of its carboxyl group. a** AFM-IR spectrum of a bexarotene (Bex, green) film deposited on a gold substrate. We represent two spectra: (i) the average + SE of the raw spectra ($n = 12$), and (ii) the average + SE of the preprocessed spectra ($n = 12$) by applying an adjacent averaging filter (5 pts) and a Savitzky–Golay filter (second order, 15 pts). Five different raw spectra at each position (coloured crosses) were acquired. **b** Bexarotene alone has strong tendency to form dimers stabilised by two hydrogen bonds corresponding to a C = O IR absorption at 1690 $cm^{-1}$, compared to the 1720–1745 $cm^{-1}$ range of bexarotene forms with one hydrogen bond (Supplementary Fig. 9)[47]. **c** Aβ42 fibrils show the typical spectrum of proteins (blue), while fibrils incubated with bexarotene (red) show an additional chemical signature related to the small molecule and with a frequency of absorption at ~1725 $cm^{-1}$, demonstrating that bexarotene is in the monomeric form and interacting with the fibrils through a single hydrogen bonding (red or orange). **d** Schematic of the fibril (blue) and bexarotene (green stars) interaction through hydrogen bonding. When bonded with the fibril or oligomer, bexarotene is likely to assume a conformation such that its bulky part is projected away from the nearest amino acid side chains.

separation inside between the spectra sub-groups. The first two PCs were sufficient to generalise the spectral differences (>80% spectral difference). A negative PC1 and PC2 were correlated with the chemical signature of the characteristic amide bands I (1700–1600 $cm^{-1}$) and II (1580–1500 $cm^{-1}$) of protein, dominating the spectra in the absence of bexarotene. While positive PC1 and PC2 were correlated with the chemical signature of bexarotene at 1700–1750 $cm^{-1}$ (C=O), 1580 and 1450 $cm^{-1}$ (C–O) (Fig. 4a). As expected, the spectra of the aggregates in presence of bexarotene (red) had mixed grouping properties when compared with the IR absorption signature of the small molecule (green) and the aggregates alone (blue). Thus, the analysis demonstrates with high statistical significance the presence of the chemical signature of bexarotene within the Aβ42 aggregates incubated with the molecule.

**Bexarotene interacts with Aβ42 aggregates through H-bonding.** The increased IR absorption in the spectra of the aggregates incubated with bexarotene showed two main peaks of absorption at 1750–1700 and 1450 $cm^{-1}$ (Fig. 3c, d), related to the vibrations of the C=O and C–O bonds of the small molecule alone (Fig. 4a). Bexarotene alone in solution has strong tendency to be in a dimeric form through a bidentate coordination (Fig. 4b), thus showing a typical frequency of the C=O bond at

approximately 1690 $cm^{-1}$, as previously shown in literature in the similar case of benzoic acid and its derivatives[46]. As shown by the average spectra in Fig. 4c and in the loading plots of PCA (red arrows in Fig. 3g and Supplementary Fig. 8), the C=O peak of bexarotene in the presence of the aggregates is significantly shifted to higher wavenumbers (~1730 $cm^{-1}$). This chemical shift demonstrates that bexarotene is transitioning from a dimeric form with double coordination to a monomeric form coordinating with the amyloid aggregates through hydrogen bonding, thus causing the shift of the carbonyl peak of bexarotene to higher energies (Supplementary Fig. 9)[47]. This difference is also demonstrated by comparing the experimental spectra of the fibrillar aggregates incubated with bexarotene and the calculated spectrum obtained summing the spectroscopic response of the fibrils and the small molecule alone (Supplementary Figs. 9 and 10). The two spectra differ significantly in the region of the carbonyl group of the carboxyl group and the N–H bond of the protein, which are involved in hydrogen bonding, thus demonstrating that these chemical bonds are involved in the protein–small molecule interaction. The main possible mechanisms of hydrogen bonding (Supplementary Fig. 9) is the direct interaction of the carboxyl group with the protein backbone (Fig. 4d). The C=O bond of the carboxyl group of the small molecule can form a hydrogen bond with the N–H of the protein backbone (red circle in Fig. 4d) or the OH group of the molecule

can form a hydrogen bond with the C=O group of the protein backbone (orange circle in Fig. 4d). Further possible interactions would be hydrogen bonding with the charged amino acids side chains of the Aβ42 protein, such as, glutamic acid, aspartic acid, arginine, lysine and histidine[48]. Other possible interactions between the carboxyl group and the protein are disfavoured due to molecular bulkiness, steric hindrance and the relatively long distance between two consecutive parallel amide hydrogens in the polypeptide chain.

In summary, the analysis of the spectra demonstrates that the hydrophilic moiety of bexarotene, specifically its carboxyl group, binds through hydrogen bonding to Aβ42 aggregates (Fig. 4d). Since the bexarotene derivative containing a methyl ester group does not modify the kinetics of Aβ42 aggregation, these results indicate that inhibiting aggregation requires the stronger and longer range hydrogen bonding of the carboxyl group over the weaker hydrogen bond of the methyl ester group (Fig. 1c)[49]. Furthermore, the inability of the bexarotene methoxy derivative to inhibit the aggregation pathway of Aβ42 may be related to other differences with the bexarotene molecule, such as the presence of the methyl group preventing the C=O interaction and a different ionisation state (pKa) of the carboxyl group in these two molecules.

## Discussion

In this study, we applied IR nanospectroscopy in combination with multivariate data analysis to first study the secondary structure of Aβ42 oligomeric and fibrillar amyloid species at the single aggregate scale, and then prove their chemical interactions with a small-molecule inhibitor of Aβ42 aggregation with single bond resolution. Initially, we characterised the aggregates of Aβ42, proving a transition from antiparallel to parallel intermolecular β-sheet content when going from the oligomeric to the fibrillar state. Then, we demonstrated that the small molecule binds through its hydrophilic moiety to Aβ42 aggregates mainly through a hydrogen bond made by its carboxyl group. These results demonstrate that IR nanospectroscopy enables the identification at atomic level of the interactions between a small molecule and its target, which has been to date hindered for Aβ42 aggregates due their transient and heterogeneous nature. We therefore anticipate that the use of IR nanospectroscopy in drug discovery will open the way to systematic structure–activity relationship studies in compound optimisation campaigns.

In conclusion, this work establishes IR nanospectroscopy as a powerful tool for characterising interactions of drugs with protein aggregates at the single oligomer level, and identifying key interaction signatures. More broadly, the development of single-molecule biophysical methodologies represents a fruitful avenue to address the challenge of unravelling the interaction of therapeutic compounds with aggregating proteins, such as small molecules, as well as antibodies. Such a molecular level understanding is key for establishing rational approaches to prevention and design pharmacological approaches to misfolding diseases.

## Methods

**Aβ peptides**. The recombinant Aβ(M1–42) peptide (MDAEFRHDSGY EVHHQ KLVFF AEDVGSNKGA IIGLMVGGVV IA), here called Aβ42, was expressed in the *Escherichia coli* BL21 Gold (DE3) strain (Stratagene, CA, USA) and purified[39]. The purification procedure involved sonication of *E. coli* cells, dissolution of inclusion bodies in 8 M urea, and ion exchange in batch mode on diethylaminoethyl cellulose resin and lyophilization. The lyophilised fractions were further purified using Superdex 75 HR 26/60 column (GE Healthcare, Buckinghamshire, UK) and eluates were analysed using SDS-PAGE for the presence of the desired protein product. The fractions containing the recombinant protein were combined, frozen using liquid nitrogen and lyophilised again. Bexarotene was obtained from Sigma-Aldrich, while the derivative of bexarotene was obtained from ArkPharm and both were of the highest purity available.

**Sample preparation for kinetic experiments**. Solutions of monomeric peptides were prepared by dissolving the lyophilised Aβ42 peptide in 6 M GuHCl. Monomeric forms were purified from potential oligomeric species and salt using a Superdex 75 10/300 GL column (GE Healthcare) at a flowrate of 0.5 ml/min, and were eluted in 20 mM sodium phosphate buffer, pH 8 supplemented with 200 μM EDTA and 0,02% NaN₃. The centre of the peak was collected and the peptide concentration was determined from the absorbance of the integrated peak area using $\varepsilon_{280} = 1490$ l/mol cm. The obtained monomer was diluted with buffer to the desired concentration and supplemented with 20 μM Thioflavin T (ThT) from a 2 mM stock. All samples were prepared in low-binding eppendorf tubes on ice using careful pipetting to avoid introduction of air bubbles. Each sample was then pipetted into multiple wells of a 96-well half-area, low-binding, clear bottom and PEG-coating plate (Corning 3881), 80 μl per well, in the absence and the presence of bexarotene.

**Kinetic assays**. Assays were initiated by placing the 96-well plate at 37 °C under quiescent conditions in a plate reader (Fluostar Omega, Fluostar Optima or Fluostar Galaxy, software version 5.4, BMGLabtech, Offenburg, Germany). The ThT fluorescence was measured through the bottom of the plate with a 440 nm excitation filter and a 480 nm emission filter. The ThT fluorescence was followed for three repeats of each sample, which were fully consistent and reproducible.

**Theoretical analysis**. The time evolution of the total fibril mass concentration, $M$ (t), is described by the following integrated rate law[22,50]:

$$\frac{M(t)}{M(\infty)} = 1 - \alpha\left(\frac{B_+ + C_+}{B_+ + C_+ e^{kt}} \frac{B_- + C_+ e^{kt}}{B_- + C_+}\right)^{\frac{k_\infty^2}{kk_\infty}} e^{-k_\infty t} \tag{1}$$

To capture the complete assembly process, only two particular combinations of the rate constants define most of the macroscopic behaviours. These are related to the rate of formation of new aggregates through primary pathways $\lambda = \sqrt{2k_+ k_n m(0)^{n_c}}$ and secondary pathways $\kappa = \sqrt{2k_+ k_2 m(0)^{n_2+1}}$, where the initial concentration of soluble monomers is denoted by $m(0)$, $n_c$ and $n_2$ describe the dependencies of the primary and secondary pathways on the monomer concentration, and $k_n$, $k_+$ and $k_2$ are the rate constants of the primary nucleation, elongation and secondary nucleation, respectively. Using Eq. (1), the experimental data of the aggregation of Aβ42 in the absence and presence of bexarotene can be described via a concomitant decrease in both $k_n$ and $k_2$.

**FTIR measurements**. Attenuated total reflection infrared spectroscopy (ATR-FTIR) was performed using a Bruker Vertex 70 spectrometer equipped with a diamond ATR element and spectra were collected by the built-in software OPUS (version 8.2, Bruker, USA). Two micromolar samples of Aβ42 aggregates in the absence and presence of threefold excess were centrifuged and re-suspended in buffer to a final concentration of 2 mM. Spectra were acquired with a resolution of 4 cm⁻¹ and all spectra were processed using Origin Pro 2019 software. The spectra were averaged (3 spectra with 512 co-averages), smoothed applying a Savitzky–Golay filter (second order, 9 pts). The FTIR measurements were performed for three independent samples, which were fully consistent and reproducible.

**Atomic force microscopy**. AFM was performed on positive functionalized mica substrates. We cleaved the mica surface and we incubated it for 1 min with 10 μl of 0.5% (v/v) (3-aminopropyl)triethoxysilane (APTES, from Sigma) in Milli-Q water. Then, the substrate was rinsed three times with 1 ml of Milli-Q water and dried by gentle stream of nitrogen gas. Finally, for each sample, an aliquot of 10 μl of the solution was deposited on the functionalized surface. The droplet was incubated for 10 min, then rinsed by 1 ml of Milli-Q water and dried by the gentle stream of nitrogen gas. The preparation was carried out at room temperature. AFM maps were acquired by means of an NX10 and its built-in software XEI (version 4.3.4, Park systems, South Korea) operating in tapping mode and equipped with a silicon tip (μmasch, 2 N/m) with a nominal radius of ~8 nm. Image flattening and single aggregate cross-sectional dimension analysis were performed by SPIP (Version 7.3.4, Image Metrology) software.

**AFM-IR measurements, maps treatment and analysis**. Analysis by nanoIR2 (Anasys Instrument, USA) was performed on atomically flat and positive gold substrates with a nominal roughness of 0.36 nm (Platypus Technologies, USA)[51]. The root mean square roughness of the AFM maps was measured by SPIP (Image metrology, Denmark). To prepare the protein samples, an aliquot of 10 μl of sample was deposited on the flat gold surface for 30 s, to reduce mass transport phenomena during drying. Successively, the droplet was rinsed by 1 ml of Milli-Q water and dried by a gentle stream of nitrogen. The morphology of the protein samples was scanned by the nanoIR microscopy system, with a rate line within 0.1–0.5 Hz and in contact mode. All AFM maps were acquired with a resolution between 2 and 10 nm/pixel. A silicon gold-coated PR-EX-nIR2 (Anasys, USA) cantilever with a nominal radius of ~30 nm and an elastic constant of about 0.2 N/m was used. To use gold–gold rod-like antenna effect the IR

light was polarised perpendicular to the surface of deposition. The AFM images were treated and analysed using SPIP software. The height images were first and second order flattened. Spectra were collected with a laser wavelength sampling of $2\,cm^{-1}$ with a spectral resolution of $\sim 1\,cm^{-1}$ and 256 co-averages, within the range $1400-1800\,cm^{-1}$. The spectra were acquired at a speed of $60\,cm^{-1}\,s^{-1}$. All spectra and maps were acquired at the same power of background laser power, between approximately 0.35 and 1 mW and with a pulse width between 40 and 100 ns. Since the spectral background line shape slightly depends on laser power, the spectra were normalised by the QCL emission profile by the built-in Analysis Studio software (version 3.15). The chemical maps and spectra were acquired on randomly chosen aggregates on each sample's surface. All measurements were performed at room temperature under controlled nitrogen atmosphere with residual real humidity below 5%. The spectra were acquired by using measuring off-resonance by ORS-nanoIR, on the left of the IR amplitude maximum resonant frequency[36,52,53].

**Spectra analysis and statistical significance**. Spectra were analysed using the microscope's built-in Analysis Studio (version 3.15, Anasys, USA) and OriginPRO. On each protein aggregate with/without small molecule at least 3 AFM-IR spectra were co-averaged, in order to have a standard error of the average in the order 5%. The spectra where despiked, an adjacent averaging filter (5 pts) and a Savitzky–Golay filter (15 pts, second order) were applied in series and then the spectra were normalised to one respect to the maximum of intensity. The structural contributions were investigated by second derivative analysis to deconvolution of the amide band I[25,27,54]. The second derivatives were smoothed by a Savitzky–Golay filter (second order, 7 pts).

For the PCA, the spectra were smoothed and normalised spectra, as described above, and further baselined to zero in the range $1400-1800\,cm^{-1}$. PCA was performed by means of OriginPRO 2019 by considering a mean-centred covariance matrix. A total of 71 spectra was sufficient to detect the subgrouping of the spectra of the aggregates incubated with/without the small-molecule bexarotene, out of the 90% confidence of the variance of each group. In all cases the first three PCs accounted for more than 85% of spectral variance. AFM-IR measurements were performed on five independent samples; several independent aggregates were measured (oligomers $n = 8$ and fibrils $n = 20$).

**Reporting summary**. Further information on research design is available in the Nature Research Reporting Summary linked to this article.

## Data availability

All data needed to evaluate the conclusions of the paper are present in the paper and the Supplementary Information file. Other data are available from the corresponding author upon request. Source data are provided with this paper.

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

## Acknowledgements

We thank Darwin College and Swiss National Fondation for Science (SNF) for the financial support (grant number P2ELP2_162116 and P300P2_171219). The research leading to these results has received funding from the Wellcome Trust and the European Research Council under the European Union's Seventh Framework Programme (FP7/2007-2013) through the ERC grant PhysProt (agreement no. 337969).

## Author contributions

F.S.R. and J.H. conceived the project. F.S.R., J.H. and S.C. performed the experiments. F.S.R. and R.H. analysed the data. F.S.R., M.V. and T.P.J.K. wrote and commented the article.

## Competing interests

The authors declare no competing interests.
