## [Peer Review File · Nature Communications]

REVIEWER COMMENTS

Reviewer #1 (Remarks to the Author):

Excellent paper describing important work to develop AFM-FTIR to study the interaction of amyloid aggregates with modifier compounds. This has important implications for drug development as well as mechanistic studies of protein aggregation in the long run. I really like technique and the analysis (such as the use of PCA etc), and the care with which it is presented here. The main claims of this paper are that the interaction between a known inhibitor of amyloid aggregation and the amyloids can be mapped and understood in mechanistic detail. I think this claim is well substantiated, but this has also to do with the fact that this molecule only postpones aggregation mildly, and so the complex still forms. More potent molecules would not allow the analysis presented.

Questions:

1. The bexarotene derivative is a bit of a deus-ex-machina but essential element in the reasoning: what was the rationale for using this particular one? serendipitous find? Like it is presented now it is a bit of retrofit to the AFM data? I get the feeling many derivatives have been studied?
2. They did excellent work by comparing the spectral signature of the molecule bound and unbound. There is very little said about the fibrils themselves. To me it seems that the peaks referring to the fibril structure when bound and unbound (amide I-II) to the molecule are very similar, suggesting there is minimal changes in the morphology of the fibrils. It would be great if they could show this more clearly, potentially with some high res AFM images and comment on this.
3. Most spectral measurements are performed in midsection of grown fibrils. These results clearly show the blocking of secondary nucleation that is templated on this surface. Was it possible to check the spectral signatures of single fibrils at the tips? And if so, where there other changes seen there?

Minor suggestion:

It is always easier when the second derivative analysis is scaled to the actual spectra. This way the reader can assign the minima easier.

Joost Schymkowitz

Reviewer #2 (Remarks to the Author):

Knowles and coworkers sought to characterize structural elements of single A β 42 oligomers and fibrils using AFM in conjunction with infrared (IR) nanospectroscopy, and the interaction with bexarotene, an aggregation inhibitor.

The AFM-IR technique is exciting and has significant potential for elucidating physical interactions between biomolecules. The data is impressive and presented well. The PCA strengthens the overall findings and helps to identify the molecular interactions.

There are some concerns with the data and how it relates to previous findings, the author's interpretations of the results, and whether the impact is appropriate for the current journal. These concerns are presented below.

1. It is surprising and somewhat puzzling that the A β 42 fibrils retain the antiparallel β -sheet signature observed in the oligomers (and to the same extent). This antiparallel structure is not really observed in other molecular-level structural studies of full-length A β 42 fibrils. The authors should discuss this contradiction. It may be possible that both species are present in the AFM sample.

2. The ThT data clearly shows that bexarotene inhibits A β 42 aggregation, while the methoxy derivative does not. Subsequent findings implicated the C=O bond in the bexarotene interaction with A β 42 oligomers and fibrils, which was ascribed to H-bonding with an A β 42 N-H. The problem is that the derivative should still be able to participate in the same H-bonding interaction. It would only be restricted in its ionization properties. The authors need to clarify this. As it stands, the overall picture does not completely come together.

3. This reviewer is enthusiastic about the potential for the IR nanospectroscopy technique, but not fully convinced the impact of the findings are at a high enough level for this journal. The A β 42 structural findings are somewhat contradictory and not novel. The inhibitor interaction can be inferred, but perhaps not measured as carefully, by other strategies.

Reviewer #3 (Remarks to the Author):

The manuscript entitled "Infrared Nanospectroscopy Reveals the Molecular Interaction Fingerprint 2 of an Aggregation Inhibitor with Single A β 42 Oligomers" by Ruggeri and co-workers reports nanoscale structural characterization of inhibition of A β 42 aggregation by bexarotene, an anticancer drug. The researchers first examined structural differences between A β 42 oligomers and fibrils using atomic force microscopy infrared nanospectroscopy (AFM-IR). Next, the researchers probed kinetics of A β 42 aggregation with and without bexarotene. Their ThT findings demonstrated that bexarotene causes dose-dependent inhibition of the peptide aggregation. Lastly, the researchers probed structural organization of A β 42 aggregates formed in the presence of bexarotene. Their findings demonstrate that bexarotene interacts with the peptide via hydrogen bonding. Thus, the aggregates grown in the presence of bexarotene contain this molecule in their structure. The use of chemometrics further proven this experimental evidence.

The manuscript is well-written and all reported results are clearly presented and extensively discussed. The manuscript can be strengthening up by investigation of toxicity of A β 42-bexarotene aggregates. The reviewer encourages the researchers to consider such an experiment in their future work.

Reviewer 1

Reviewer: Excellent paper describing important work to develop AFM-FTIR to study the interaction of amyloid aggregates with modifier compounds. This has important implications for drug development as well as mechanistic studies of protein aggregation in the long run. I really like technique and the analysis (such as the use of PCA etc), and the care with which it is presented here.

Answer: We would like to thank the reviewer for these very positive comments on the quality and significance of our study and methods for understanding protein aggregation and for drug development.

Reviewer: The main claims of this paper are that the interaction between a known inhibitor of amyloid aggregation and the amyloids can be mapped and understood in mechanistic detail. I think this claim is well substantiated, but this has also to do with the fact that this molecule only postpones aggregation mildly, and so the complex still forms. More potent molecules would not allow the analysis presented.

Answer: We thank the reviewer for raising this point that stresses the relevance of the capabilities of our single molecule chemical approach to study the interaction of single fibrils and especially oligomers already with mildly potent small molecules, as we demonstrate in **Figure 2-3**. Molecules that are more potent would further delay the aggregation or redirect it at a stage where only oligomeric populations may be present. Although the number of the oligomeric aggregates may further decrease, these numbers are still likely to be high when performing measurements at the single molecule level. Furthermore, the stronger affinity and/or chemical bonding of a potential more potent drug with the oligomers would in turn facilitate the characterisation of their interactions.

Reviewer: 1. The bexarotene derivative is a bit of a deus-ex-machina but essential element in the reasoning: what was the rationale for using this particular one? serendipitous find? Like it is presented now it is a bit of retrofit to the AFM data? I get the feeling many derivatives have been studied?

Answer: Following the questions of the reviewer, we have clarified these points in the manuscript. To identify bexarotene, we used a rational drug discovery strategy against A β 42 aggregation based on chemical kinetics (*Habchi, Science Advances, 2016*). Then, to identify the derivative, we studied systematically how different modifications of the hydrophilic and hydrophobic parts of the small molecules could affect the kinetics of aggregation (*Chia, PNAS, 2018*).

Reviewer: 2. They did excellent work by comparing the spectral signature of the molecule bound and unbound. There is very little said about the fibrils themselves. To me it seems that the peaks referring to the fibril structure when bound and unbound (amide I-II) to the molecule are very similar, suggesting there is minimal changes in the morphology of the fibrils. It would be great if they could show this more clearly, potentially with some high res AFM images and comment on this.

Answer: We thank the reviewer for this useful suggestion. In **Supplementary Figure 5-6**, we compare the morphology of the bound/unbound state of the fibrils, which is indeed very similar. In the **new Supplementary Figure 6** added in response to the referee's helpful comments, we now compare the spectra of the bound/unbound fibrils. As the reviewer noticed, the spectra and their Amide I bands show only minor differences, which could be also associated to the presence of residual IR absorption of bexarotene in the amide I of bound fibrils. Thus, overall, there are only minor structural differences between bound/unbound fibrils. We have clarified this point in the revised manuscript.

Reviewer: 3. Most spectral measurements are performed in midsection of grown fibrils. These results clearly show the blocking of secondary nucleation that is templated on this surface. Was it possible to check the spectral signatures of single fibrils at the tips? And if so, where there other changes seen there?

Answer: For the compounds studied on our paper the evidence from kinetic studies is that processes, such as elongation, that occur at the end of the fibrils are not affected. Thus, we have focused our studies on the midsection of the fibrils. Future enhancements to the sensitivity and resolution of the technique may allow further spatial resolution to be obtained for small molecule interactions along the lines suggested by the referee.

Reviewer: Minor suggestion: It is always easier when the second derivative analysis is scaled to the actual spectra. This way the reader can assign the minima easier.

Answer: Following the suggestion of the reviewer, we have now rescaled the spectra and their second derivative to the same range in the **new Supplementary Figure 10**.

Reviewer 2

Reviewer: Knowles and coworkers sought to characterize structural elements of single A β 42 oligomers and fibrils using AFM in conjunction with infrared (IR) nanospectroscopy, and the interaction with bexarotene, an aggregation inhibitor. The AFM-IR technique is exciting and has significant potential for elucidating physical interactions between biomolecules. The data is impressive and presented well. The PCA strengthens the overall findings and helps to identify the molecular interactions.

Answer: We are grateful to the reviewer for these very positive comments and for highlighting the importance and broad impact of our AFM-IR approach.

Reviewer: There are some concerns with the data and how it relates to previous findings, the author's interpretations of the results, and whether the impact is appropriate for the current journal. These concerns are presented below.

Answer: Following the useful comments and suggestions of the reviewer, as detailed below, we have updated the discussion of the data in the manuscript, which we hope has further enhanced clarity.

Question 1: It is surprising and somewhat puzzling that the A β 42 fibrils retain the antiparallel β -sheet signature observed in the oligomers (and to the same extent). This antiparallel structure is not really observed in other molecular-level structural studies of full-length A β 42 fibrils. The authors should discuss this contradiction. It may be possible that both species are present in the AFM sample.

Answer: We thank the reviewer for this comment that provides us with the opportunity to further clarify the discussion of the structural conformation of the oligomers and fibrils. Our analysis shows that our single molecule data are in agreement with results from previous structural studies performed on bulk samples. It is widely accepted in literature that the presence in the IR spectra of amyloid fibrils of both a high frequency (1690 cm^{-1}) and a low frequency (1620 cm^{-1}) band of intermolecular β -sheet structure is fully compatible with a parallel β -sheet structure. Indeed, while the peptides have parallel β -sheet conformation, some residues have Ramachandran angles and IR spectra causing the presence of an antiparallel β -sheet component, which is routinely observed for A β fibrils in bulk by FTIR (Zanjani, *Biophysical Journal*, 2020; Habchi, *Nature Chemistry*, 2018; Okada, *ACS Chemical Neuroscience*, 2019; Moran, *The Journal of Physical Chemistry Letters*, 2014). By contrast, the presence of only the high-frequency band is strictly associated to antiparallel β -sheet conformation.

Question 2. The ThT data clearly shows that bexarotene inhibits A β 42 aggregation, while the methoxy derivative does not. Subsequent findings implicated the C=O bond in the bexarotene interaction with A β 42 oligomers and fibrils, which was ascribed to H-bonding with an A β 42 N-H. The problem is that the derivative should still be able to participate in the same H-bonding interaction. It would only be restricted in its ionization properties. The authors need to clarify this. As it stands, the overall picture does not completely come together.

Answer: We thank the reviewer for his insightful comment, which enabled us to clarify the section of the manuscript regarding the small molecules-aggregates interaction. As the reviewer has rightly pointed out, the chemical difference between bexarotene and its derivative is in their ionisation properties and hydrogen bonding capabilities. Specifically, bexarotene contains a carboxyl group, while the derivative has a methyl ester group in its location. The carboxyl group can form a stronger H-bond than the methyl ester group (Patrick, *An introduction to medicinal chemistry. 6th edition. Oxford university press, 2017*). Furthermore, the carboxyl group has multiple options to create an H-bond with the protein backbone (C=O and N-H) when compared to the ester (only N-H). We have now clarified this point in the text of the manuscript and the **new Figure 4**.

Question 3: This reviewer is enthusiastic about the potential for the IR nanospectroscopy technique, but not fully convinced the impact of the findings are at a high enough level for this journal. The A β 42 structural findings are somewhat contradictory and not novel. The inhibitor interaction can be inferred, but perhaps not measured as carefully, by other strategies.

Answer: We thank the reviewer for the positive evaluation of our AFM-IR approach. Here, the unprecedented sensitivity of AFM-IR has enabled us to characterise the interactions of a small molecule with aggregating peptide systems. Due to the heterogeneity of such systems, the detailed picture only emerges on the single molecule level. These results establish AFM-IR as a new powerful tool in biosciences to study the interaction of drugs with heterogeneous targets at the single molecule scale. As the referee points out, some of these findings can be inferred from bulk data and in such cases our structural findings are in excellent agreement with previous reports in literature. The unrivalled sensitivity has also enabled us to acquire for the first time

the chemical signature and evaluate the secondary structure of oligomeric species of A β 42 at the single molecule scale. These oligomers have dimensions (~5 nm) thus the assessment of their secondary structure was out of the sensitivity of detection of conventional structural bulk methods and the AFM-IR technique until now.

Reviewer 3

Reviewer: The manuscript entitled “Infrared Nanospectroscopy Reveals the Molecular Interaction Fingerprint of an Aggregation Inhibitor with Single A β 42 Oligomers” by Ruggeri and co-workers reports nanoscale structural characterization of inhibition of A β 42 aggregation by bexarotene, an anticancer drug. The researchers first examined structural differences between A β 42 oligomers and fibrils using atomic force microscopy infrared nanospectroscopy (AFM-IR). Next, the researchers probed kinetics of A β 42 aggregation with and without bexarotene. Their ThT findings demonstrated that bexarotene causes dose-dependent inhibition of the peptide aggregation. Lastly, the researchers probed structural organization of A β 42 aggregates formed in the presence of bexarotene. Their findings demonstrate that bexarotene interacts with the peptide via hydrogen bonding. Thus, the aggregates grown in the presence of bexarotene contain this molecule in their structure. The use of chemometrics further proven this experimental evidence. The manuscript is well-written and all reported results are clearly presented and extensively discussed.

Answer: We would like to thank the reviewer for these very positive comments on our work.

Reviewer: The manuscript can be strengthening up by investigation of toxicity of A β 42-bexarotene aggregates. The reviewer encourages the researchers to consider such an experiment in their future work.

Answer: The interest in studying the molecular interaction of bexarotene and A β 42 arises from previous results showing that bexarotene is capable of clearing A β aggregates and reverse neuronal deficit in mice (*Cramer, Science, 2012*), as well as delay the formation of A β aggregates and its related muscular paralysis in *C. Elegans* models of A β 42-mediated dysfunction (*Habchi, Science Advances, 2016*). We thank the reviewer for allowing us to expand on this point, which we clarified in the introduction of the manuscript.

REVIEWERS' COMMENTS

Reviewer #1 (Remarks to the Author):

All my comments have been adequately addressed

Reviewer #2 (Remarks to the Author):

The authors have carefully addressed my questions regarding the interpretation of the results in this well-done and high impact study.

IR nanospectroscopic analysis and interpretation of the A β transition from monomer to oligomers to fibrils has been clarified, provides significant future potential, and is now much more in line with other structural observations.

The IR nanospectroscopy data clearly show the signature C=O & C-O bexarotene peaks in the A β aggregates, but this result does not definitively demonstrate the exact nature and location of the interaction. I do feel that the bulk of the findings support an H-bonding interaction between A β and bexarotene that is abrogated in some manner with the methoxy derivative. The authors have now generalized the interaction to the bexarotene carboxyl group (C=O & C-O) and this is probably the best that can be surmised at this point. It should be noted that there is no IR nanospectroscopy data presented for the bexarotene methoxy derivative to rule out an interaction with A β .

For conversation sake, since both compounds have a carbonyl (C=O) available for H-bonding, there are other considerations such as the ionization state (pKa) of the bexarotene carboxyl group and the possibility that there is no interaction between C-OH and A β but that the methyl group on the bexarotene methoxy derivative disrupts/prevents the C=O interaction in some manner.

I recommend publication after consideration of the aforementioned points.

Reviewer #3 (Remarks to the Author):

the authors addressed all reviewer's questions and I suggest to accept this manuscript for the publication.

Reviewer 1

Reviewer: All my comments have been adequately addressed

Answer: We would like to thank the reviewer for all her/his useful comments and suggestions.

Reviewer 2

Reviewer: The authors have carefully addressed my questions regarding the interpretation of the results in this well-done and high impact study.

Answer: We would like to thank the reviewer for the very positive evaluation of our study.

Reviewer: The IR nanospectroscopy data clearly show the signature C=O & C-O bexarotene peaks in the A β aggregates, but this result does not definitively demonstrate the exact nature and location of the interaction. I do feel that the bulk of the findings support an H-bonding interaction between A β and bexarotene that is abrogated in some manner with the methoxy derivative. The authors have now generalized the interaction to the bexarotene carboxyl group (C=O & C-O) and this is probably the best that can be surmised at this point.

Answer: We fully agree with the reviewer, our results successfully demonstrate the interaction of bexarotene with A β aggregates by hydrogen bonding. In line with the suggestion of the reviewer, we decided to generalise the interaction since multiple locations and nature of the bond could be possible.

Reviewer: It should be noted that there is no IR nanospectroscopy data presented for the bexarotene methoxy derivative to rule out an interaction with A β . For conversation sake, since both compounds have a carbonyl (C=O) available for H-bonding, there are other considerations such as the ionization state (pKa) of the bexarotene carboxyl group and the possibility that there is no interaction between C-OH and A β but that the methyl group on the bexarotene methoxy derivative disrupts/prevents the C=O interaction in some manner.

Answer: Following the useful suggestion of the reviewer, we have added the following sentence to the final results section of the manuscript: *“Furthermore, the inability of the bexarotene methoxy derivative to inhibit the aggregation pathway of A β 42 may be related to other differences with the bexarotene molecule, such the presence of the methyl group preventing the C=O interaction and a different ionisation state (pKa) of the carboxyl group in these two molecules.”*

Reviewer 3

Reviewer: The authors addressed all reviewer's questions and I suggest to accept this manuscript for the publication.

Answer: We would like to thank the reviewer for all her/his useful comments and suggestions.